# Stability of Aggregates Made by Earthworms in Soils with Organic Additives

Agnieszka Józefowska [1,*] , Karolina Woźnica [1] , Justyna Sokołowska [1] , Agata Sochan [2] , Tomasz Zaleski [1] , Magdalena Ryżak [2] and Andrzej Bieganowski [2]

[1] Department of Soil Science and Agrophysics, University of Agriculture in Krakow, Al. Mickiewicza 21, 30-120 Krakow, Poland; karolinawoznica@o2.pl (K.W.); j.sokolowska123@gmail.com (J.S.); tomasz.zaleski@urk.edu.pl (T.Z.)

[2] Institute of Agrophysics PAS, Doświadczalna 4, 20-290 Lublin, Poland; a.sochan@ipan.lublin.pl (A.S.); m.ryzak@ipan.lublin.pl (M.R.); a.bieganowski@ipan.lublin.pl (A.B.)

* Correspondence: agnieszka.jozefowska@urk.edu.pl

**Abstract:** Earthworm activity is a key factor in creating soil aggregates, but introduced organic matter and abiotic factors are also equally important. The purpose of this study was to investigate the stability of aggregates made by earthworms in soils with organic additives. Additionally, the two aggregate stability measurement methods were compared: (i) the wet-sieve method and (ii) the laser diffraction method. A six-month container experiment containing sixteen treatments and controls were made. Each treatment included one of four types of soil texture: sand, loam, silty loam and clay, and one of four additives: straw, peat, compost and compost with added microorganisms. To each treatment, six earthworms were added, two each of species commonly occurring in Polish soils: *Dendrodrilus rubidus*, *Aporrectodea caliginosa* and *A. rosea*. This study confirmed that earthworm activity was the factor favoring aggregate formation. In terms of the investigated organic additives, the efficiency of aggregate creation was as follows: compost with active bacteria, compost, peat and straw. Nevertheless, earthworms alone, without the addition of an organic additive, did not form permanent aggregates. The wet sieving and laser diffractometry methods of measuring aggregate stability were comparable for silty, clayey and loamy soils.

**Keywords:** soil texture; microbial activity; compost; straw; peat



## 1. Introduction

The formation of soil aggregates is complicated and depends on the parent material as well as physical and biochemical soil-forming processes [1]. The aggregate structure is closely associated with the presence of clay minerals and organic matter [2,3]. Clay minerals are the most reactive and promote interactions with ions, mineral particles and organic matter in the soil because of their specific surface area and surface charges [4]. Chenu et al. [5], in their preliminary observations of the clay fraction, noticed that the organic constituents and organic matter were strongly associated with clay particles, which increased the hydrophobicity of clay minerals. Conversely, in soils with the dominance of sand fraction, the stability of aggregates is low [6–8]. This is because the sand has a large size and low surface area compared with clay, so its bonding capacity, e.g., with metal cations or organic molecules, is very low [9]. The differences in surface area, charge of clay and swelling behavior influence the forming process and aggregate stability [4,10–12]. Moreover, aggregate formation is a comprehensive interaction between soil minerals and soil organic matter [13]. The crucial factor in increasing aggregate stability is appropriate management of organic matter addition [14]. In many cultivation systems, fresh organic matter is periodically returned to the soil as litter or crop residues, enriching the soil in different amounts based on the quality of organic inputs [15]. Concerning soil organic matter, many authors [16–19] emphasized the positive influence of organic carbon on soil

aggregation. The influence is closely linked to the individual composition of soil organic matter, especially with polysaccharides and humic and fulvic acids. According to Tan [20] the major factors in the formation and cementation of soil structure, and thereby the soil aggregation, are humic acids, which are crucial for sandy soil. As opposed to sandy soil dominated by the coarse fractions (sand), soils with fine fractions (clay, silt) show a positive correlation with total organic carbon [21,22], which indicates that the soil rich in clay minerals can create a stable organo-mineral association with a significantly large amount of organic substances [23].

Lubbers et al. [24] emphasized that earthworms increase the amount of organic carbon in the soil and as a consequence create macroaggregates. Such macroaggregates contain particulate organic matter, fungal hyphae or roots, and afterwards, during the decomposition of these macroaggregates, the organic matter becomes more resistant to microbial attack, which favors the sequestration of organic carbon in the soil [25]. Earthworms, through feeding and burrowing, are important elements in C cycling [26]. However, the type of introduced organic matter [27,28] and abiotic factors [13] are equally important in creating stable organic–mineral components as well. Moreover, earthworms play an essential role in creating soil structure by modifying soil aggregation and porosity [29]. Soil aggregates participate in sequestration of carbon in the soil and minimize microbial decomposition of organic matter. Mustafa et al. [30] indicated soil aggregate stability and soil organic carbon stock as the most influential factors for soil organic carbon and total nitrogen sequestration. Moreover, those authors noted a higher mineralization rate in macroaggregates (> 250 μm) and suggested that smaller aggregates are more capable of protecting soil organic carbon against decomposition.

Aggregate stability may be measured by a few methods. The most common is wet sieving [8], but also rain simulation or ultrasonic vibration [31] or air bubbling after immersion [32] are used. Bieganowski et al. [33,34] proposed estimating soil aggregate stability based on the changes of the median during measurement of particle size distribution by the laser diffraction method. This method for the calculation of the aggregate stability is comparable to the water-resistant index (WRI) obtained by the wet sieving method. The laser diffraction method, due to its sensitivity (accuracy), was also recommended for aggregates that have very large WRI.

Summarizing the above review, it should be stated there is a lack of full knowledge about the aggregates created by the earthworms. Therefore, it is worth further investigating the stability of aggregates made by earthworms. This study aims to verify three hypotheses:

H1: aggregate stability is related to soil texture and organic additives applied to the soil
H2: aggregate stability is connected with carbon stabilization and microbial activity
H3: two methods of measuring aggregate stability, namely (i) the wet-sieve method and (ii) the laser diffraction method are comparable.

## 2. Materials and Methods

### 2.1. Experiment Design

The research was carried out on soils with four types of textures, i.e., sand (99% sand and 1% clay), loam (31% sand and 21% clay), silty loam (4% sand and 15% clay) and clay (4% sand and 67% clay). Soils were collected from the parent material horizon (C) from four places. The reason for sampling from horizon C was that this material was in the smallest degree exposed on surface biological activity, influencing soil aggregate creation/stability. Basic properties of the soil are presented in Table S1. For each soil type, except the control samples, one of the following additives was added: straw, peat, compost or compost amended with extra microorganisms that are commercially available cultures of microorganisms containing phototropic bacteria (EmFarmaPlus$^{TM}$, ProBiotics Poland, Warsaw) (compost$_{micro}$). The doses of each additive were chosen to obtain soil with 2% of total organic carbon, which corresponds to the average content of organic carbon in Polish soils (Table S2). The carbon content in additives and the doses are presented in Table 1S. To each container (1200 mL volume) was added 400 g of sieved, structure-less soil and

proper doses of additives; into such treatment, six earthworms were added from three different species of earthworms that commonly occur in Polish soils, i.e., two earthworms each of *Dendrodrilus rubidus* (with an average mass of $0.08 \pm 0.02$ g), *Aporrectodea caliginosa* (with an average mass of $0.62 \pm 0.15$ g) and *Aporrectodea rosea* (with an average mass of $0.15 \pm 0.06$ g). The number of earthworms corresponded to an earthworm density and biomass equal to ~340 ind.·m$^{-2}$ and ~49 g·m$^{-2}$, respectively. These values are higher than in typical European soils [21,22] but allowed a faster formation of soil aggregates. The earthworms were provided with optimal conditions throughout the experiment. Containers were maintained at 10 °C. The openings in the lids of the containers ensured a constant supply of air and a soil moisture approx. 50% of water holding capacity. Each treatment was repeated 3 times; in the whole experiment was 60 containers. After six months of the experiment, the soils were analyzed.

**Table 1.** Content of carbon and nitrogen introduced to the treatments soil ($C_{intr}$, $N_{intr}$), microbial biomass carbon and nitrogen (MBC and MBN, respectively), dissolved organic carbon and nitrogen (DOC and DON, respectively) and dehydrogenase activity (DHA) in the soils. Different lower case letters mean significant differences for soil texture; uppercase letter mean significant differences for additives.

| Properties | Sand | | | | | Loam | | | | |
|---|---|---|---|---|---|---|---|---|---|---|
| | **Control** | **Straw** | **Peat** | **Compost** | **Compost$_{micro}$** | **Control** | **Straw** | **Peat** | **Compost** | **Compost$_{micro}$** |
| $C_{intr}$ | 0 | 0.09 ± 0.02 Aab | 1.23 ± 0.51 Bab | 0.59 ± 0.13 Bab | 0.75 ± 0.48 Bab | 0 | 0.21 ± 0.09 Ab | 1.05 ± 0.27 Bb | 1.04 ± 0.13 Bb | 0.95 ± 0.25 Bb |
| $N_{intr.}$ | 0 | 0.01 ± 0.17 Aa | 43.78 ± 18.45 Ba | 57.83 ± 14.13 Ca | 73.94 ± 49.92 Ca | 0 | 2.46 ± 1.47 Aa | 38.78 ± 8.42 Ba | 97.48 ± 10.64 Ca | 62.80 ± 3.19 Ca |
| MBC | 3.7 ± 0.0 Aab | 33.3 ± 29.7 Aab | 66.6 ± 38.6 ABab | 179.7 ± 145.1 Cab | 200.9 ± 106.0 BCab | 56.5 ± 50.3 Aa | 37.2 ± 9.2 Aa | 36.8 ± 9.2 ABa | 73.2 ± 41.5 Ca | 81.1 ± 22.6 BCa |
| MBN | 0.0 ± 0.0 Aba | 0.0 ± 0.0 Aab | 7.1 ± 6.4 Aab | 18.3 ± 18.3 Bab | 31.5 ± 35.2 Aab | 3.1 ± 3.6 Aa | 3.3 ± 1.1 Aa | 3.8 ± 1.9 Aa | 4.7 ± 4.8 Ba | 16.7 ± 8.0 Aa |
| DOC | 44.5 ± 4.1 Ab | 97.1 ± 14.1 Bb | 100.6 ± 23.3 Bb | 116.9 ± 19.3 Bb | 117.9 ± 5.1 Bb | 59.2 ± 11.8 Aa | 71.7 ± 14.1 Ba | 77.5 ± 9.1 Ba | 89.1 ± 20.1 Ba | 94.2 ± 15.5 Ba |
| DON | 33.7 ± 3.6 ABa | 21.7 ± 3.1 Aa | 53.7 ± 35.4 ABa | 57.3 ± 35.4 Ca | 69.7 ± 31.0 BCa | 42.9 ± 9.5 ABab | 37.3 ± 0.6 Aab | 50.4 ± 6.4 ABab | 82.7 ± 22.6 Cab | 72.3 ± 18.0 BCab |
| DHA | 0.11 ± 0.08 Ab | 0.31 ± 0.20 Ab | 0.97 ± 0.58 Ab | 1.17 ± 0.94 Ab | 1.36 ± 0.86 Ab | 0.14 ± 0.08 Ab | 2.02 ± 2.26 Ab | 0.56 ± 0.49 Ab | 0.76 ± 0.32 Ab | 1.21 ± 1.18 Ab |

Units: $C_{intr}$ g kg$^{-1}$; $N_{intr}$, MBC, MBN, DOC and DON mg kg$^{-1}$; DHA gTPF·g$^{-1}$soil·h$^{-1}$

### 2.2. Laboratory Measurements

After six months, before removing earthworms from the containers, photographs of the bottom surface with visible earthworm traces (ET) were taken from every sample. All photos were taken with fixed camera settings and light conditions. Using ImageJ 1.50i [35], cropped photos were converted into 8-bit images and binarized, and then the percent area of ET on the photos was analyzed (Figure 1).

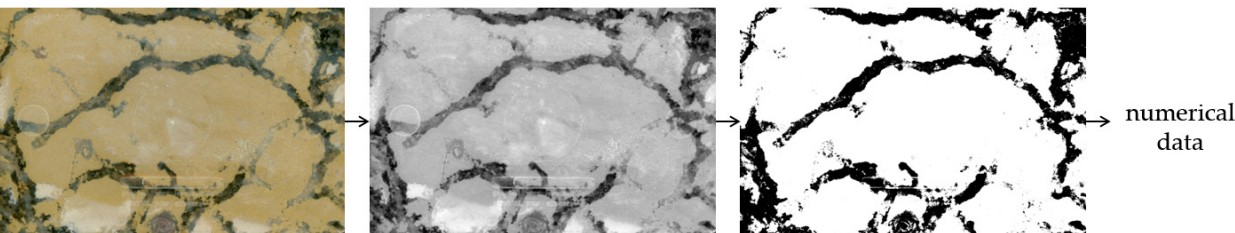

**Figure 1.** Transformation of photography to numerical data.

Fresh soil was sieved (2 mm mesh size) and stored field-moist at 4 °C for one week. Microbial biomass carbon and nitrogen (MBC and MBN, respectively) and dehydrogenase activity (DHA) were analyzed. DHA was measured using the 2,3,5-triphenyltetrazolium chloride (TTC) method, which becomes reduced to formazan (TPF) during incubation for 24 h at 37 °C [36]. MBC and MBN were determined by the fumigation–extraction method [37]. Dissolved organic carbon and nitrogen (DOC and DON, respectively) was measured using 5 mM $CaCl_2$ (soil:$CaCl_2$ ratio 1:10), filtered by 0.45 µm [38]. DOC, DON, MBC and MBN were measured using the dry combustion method with a Shimadzu TOC-VCPH TOC analyzer ((Shimadzu, Kyoto, Japan). In dry soil, the pH in 1M KCl was measured using a potentiometer (1.0:2.5, w:v, soil:water ratio), and total organic carbon ($C_{org}$) and total nitrogen ($N_t$) were measured using a LECO TruMac® CNS analyzer (LECO, St. Joseph, USA).).

Before the measurements, all aggregates were sieved through 2 mm and 1 mm sieves, and aggregates of 1–2 mm in size were analyzed. Aggregate stability was determined using the wet sieving method (WRI) using an Eijkelkamp sieving apparatus according to operating instructions (3 min ± 5 s stroke equal to 1.3 cm, at about 34 times min$^{-1}$; Eijkelkamp Agrisearch Equipment, Giesbeek, the Netherlands) [39], and the laser diffraction method using a laser diffractometer Mastersizer 2000 with Hydro G dispersion unit (Malvern, UK) [33]. The size of intact aggregates was measured using a microscope (Morphologi G3, Malvern, UK) [33]. The aggregate stability index ($ASI_{LD}$) was calculated as a slope of the straight line interpolated from two points. The first point was the median value of the aggregate size estimated based on microscope measurements. The second point was the value of the median size of aggregates after 60 s from the beginning of laser diffraction measurement (Figure 2) [33]. The direction coefficient of the straight line gives information about the stability of the aggregates. The more vertical the interpolated line, the less stable the aggregates. In the sandy texture treatments, no or few aggregates, which disintegrated during the initial process of soil treatment, were obtained. Therefore, statistical analyses relating to the stability (or water resistance) of aggregates were carried out only for soils with loamy, silty loam and clay textures.

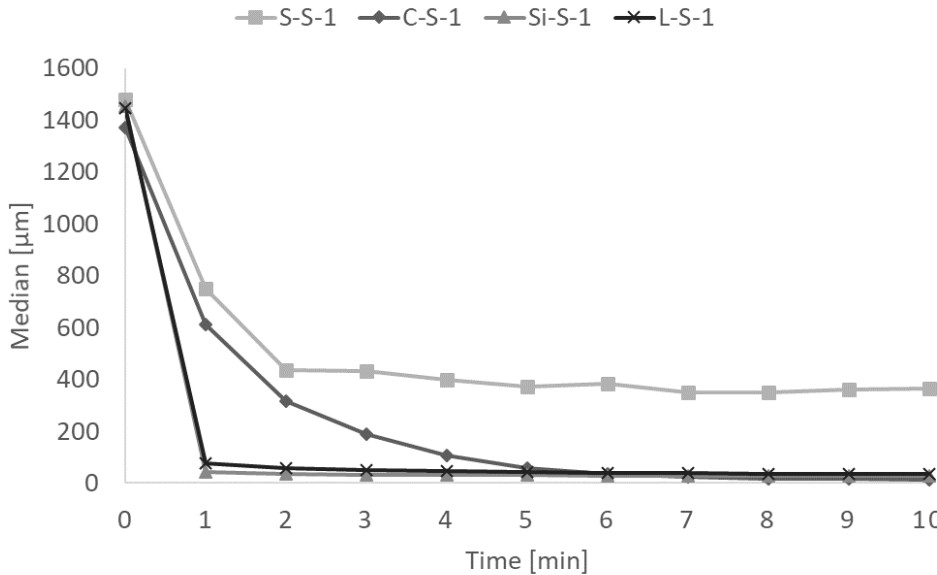

**Figure 2.** Examples of change in median particle size distribution measured by the laser diffraction method in sand (S), clay (C), silty loam (Si) and loam (L) with straw addition (S).

The amount of introduced carbon and nitrogen were quantified as total organic carbon introduced ($C_{intr}$) and total nitrogen introduced ($N_{intr}$). $C_{intr}$ was calculated according to the equation, total organic carbon content in the sample – the total organic carbon content in the control sample, and $N_{intr}$ was calculated analogously.

### 2.3. Statistical Analysis

Statistica 13.0 [40] and Canoco 5 [41] were used for statistical analyses. The Lilliefors test compliance with normal distribution was tested using Shapiro–Wilk with Lilliefors adjustment, and data without normal distribution were transformed (Log transformation formula $Y'=1 \times Y+1$). The effects of organic additives and soil texture were analyzed by a two-way ANOVA for the majority of response variables ($p = 0.05$). Tukey's HSD multiple comparison procedure was used to show the difference between treatments ($p = 0.05$). A multiple regression equation was used to investigate the relationship between the tested soil properties and the stability of aggregates; variables were selected using the stepwise method. Principle component analysis (PCA) was used to determine the main trends in the data and to indicate the approximate direction of soil variable effects and the similarities and dissimilarities between treatments.

### 3. Results

The highest amount of $C_{intr}$ was in sandy-peat and the lowest sandy soil with straw and also silty loam soil with straw. There were significant differences between $C_{intr}$ in sandy soils with peat and straw additives. In loam soils, the amount of $C_{intr}$ in treatment with straw additive was significant lower compared to treatments with peat and compost. The type of applied additive influenced the content of $N_{intr}$; treatments with compost and $compost_{micro}$ had the highest amount of introduced nitrogen compared to the straw and peat treatments. The treatments with straw characterized significant lower $N_{intr}$ than compost and $compost_{micro}$ additives in sand, loam, silty loam and clay soils. Moreover, in loam and clay soils, $N_{intr}$ content was significant lower in treatments with peat additive compared to compost. Additionally, $N_{intr}$ in silty loam–peat differed from silty loam–compost and silty loam–$compost_{micro}$. (Tables 1 and 2). The amount of introduced $C_{intr}$ was determined by the amount of organic additives added to the soils but not by the texture (Table 3).

**Table 2.** Content of carbon and nitrogen introduced to the soil ($C_{intr}$, $N_{intr}$), microbial biomass carbon and nitrogen (MBC and MBN, respectively), dissolved organic carbon and nitrogen (DOC and DON, respectively) and dehydrogenase activity (DHA) in treatment soils. Different lower case letters mean significant differences for soil texture; uppercase letter mean significant differences for additives.

| Properties | Silty Loam | | | | | Clay | | | | |
|---|---|---|---|---|---|---|---|---|---|---|
| | Control | Straw | Peat | Compost | Compost$_{micro}$ | Control | Straw | Peat | Compost | Compost$_{micro}$ |
| $C_{intr}$ | 0 | 0.04 ± 0.05 Aa | 0.57 ± 0.16 Ba | 0.78 ± 0.25 Ba | 0.66 ± 0.29 Ba | 0 | 0.33 ± 0.08 Aab | 1.01 ± 0.18 Bab | 0.95 ± 0.06 Bab | 0.49 ± 0.39 Bab |
| $N_{intr.}$ | 0 | 0.01 ± 0.15 Aa | 19.88 ± 6.99 Ba | 78.44 ± 22.68 Ca | 84.08 ± 13.44 Ca | 0 | 0.51 ± 1.20 Aa | 39.54 ± 8.17 Ba | 93.60 ± 7.43 Ca | 69.55 ± 19.65 Ca |
| MBC | 86.1 ± 67.5 Abc | 92.3 ± 3.8 Abc | 142.8 ± 49.2 ABBbc | 174.6 ± 12.2 Cbc | 129.3 ± 102.2 BCbc | 62.7 ± 13.7 Ac | 108.4 ± 57.1 Ac | 130.0 ± 76.7 aABc | 384.9 ± 9.7 Cc | 139.0 ± 34.9 BCc |
| MBN | 3.6 ± 5.5 Aa | 8.2 ± 0.9 Aa | 5.6 ± 2.8 Aa | 5.2 ± 4.9 Ba | 5.2 ± 9.8 Aa | 0.0 ± 0.1 Ab | 8.5 ± 1.9 Ab | 0.0 ± 0.0 Ab | 85.3 ± 20.2 Bb | 0.0 ± 0.0 Ab |
| DOC | 75.8 ± 9.5 Ac | 109.8 ± 31.4 Bc | 127.4 ± 15.1 Bc | 119.9 ± 6.5d Bc | 120.2 ± 12.1 Bc | 95.7 ± 3.7 Abc | 110.1 ± 9.8 Bbc | 89.4 ± 9.0 Bbc | 126.7 ± 17.2 Bbc | 109.7 ± 5.2 Bbc |
| DON | 65.2 ± 26.8 ABb | 54.4 ± 28.0 Ab | 64.1 ± 6.5 ABb | 67.1 ± 6.1 Cb | 73.6 ± 6.8 BCb | 59.4 ± 13.6 ABab | 38.0 ± 1.0 Aab | 38.0 ± 4.1 ABab | 84.8 ± 21.9 Cab | 62.5 ± 2.2 BCab |
| DHA | 0.67 ± 0.12 Aab | 0.57 ± 0.21 Aab | 0.94 ± 0.26 Aab | 0.15 ± 0.08 Aab | 0.48 ± 0.08 Aab | 0.12 ± 0.12 Aa | 0.17 ± 0.05 Aa | 0.94 ± 0.24 Aa | 0.11 ± 0.01 Aa | 0.08 ± 0.09 Aa |

Units: $C_{intr}$ g kg$^{-1}$; $N_{intr}$, MBC, MBN, DOC and DON mg kg$^{-1}$; DHA gTPF·g$^{-1}$soil·h$^{-1}$

**Table 3.** The interaction of texture and organic additives (two-way analysis of variance (ANOVA), significance level of $P = 0.05$, only significant values included). Abbreviations: WRI—water-resistant index, ASI$_{LD}$—aggregate stability index, see Tables 1 and 2.

| Properties | Interaction Additives | | Interaction Texture | | Interaction Texture × Additives | |
|---|---|---|---|---|---|---|
| | F | P | F | p | F | p |
| $C_{intr.}$ | 23.10 | 0.000 | 2.90 | 0.053 | 1.70 | 0.119 |
| $N_{intr}$ | 59.60 | 0.000 | 0.50 | 0.657 | 1.70 | 0.138 |
| MBC | 13.06 | 0.000 | 9.17 | 0.000 | 3.17 | 0.003 |
| MBN | 12.55 | 0.000 | 4.88 | 0.006 | 10.03 | 0.000 |
| DOC | 17.58 | 0.000 | 14.68 | 0.000 | 2.28 | 0.026 |
| DON | 9.02 | 0.000 | 2.77 | 0.054 | 1.17 | 0.339 |
| DHA | 1.21 | 0.323 | 4.16 | 0.01 | 1.57 | 0.140 |
| ET | 24.23 | 0.000 | 12.12 | 0.000 | 11.34 | 0.000 |
| WRI* | 7.00 | 0.000 | 940.10 | 0.000 | 1.80 | 0.126 |
| ASI$_{LD}$* | 1.00 | 0.441 | 2264.4 | 0.000 | 1.40 | 0.218 |

* without aggregates with sandy texture.

Traces of earthworms (ET) were present in from 1.9% to 59.2% of the area in the sand control and sand–compost$_{micro}$ treatments, respectively. The factors differentiating ET were the texture and the added additive and the interaction between the two factors (Figure 3a–c). Considering ET by texture, sandy soils characterized significantly the highest ET compared with the others, whereas ET in soils with straw additive was significantly higher than in treatments with compost and peat.

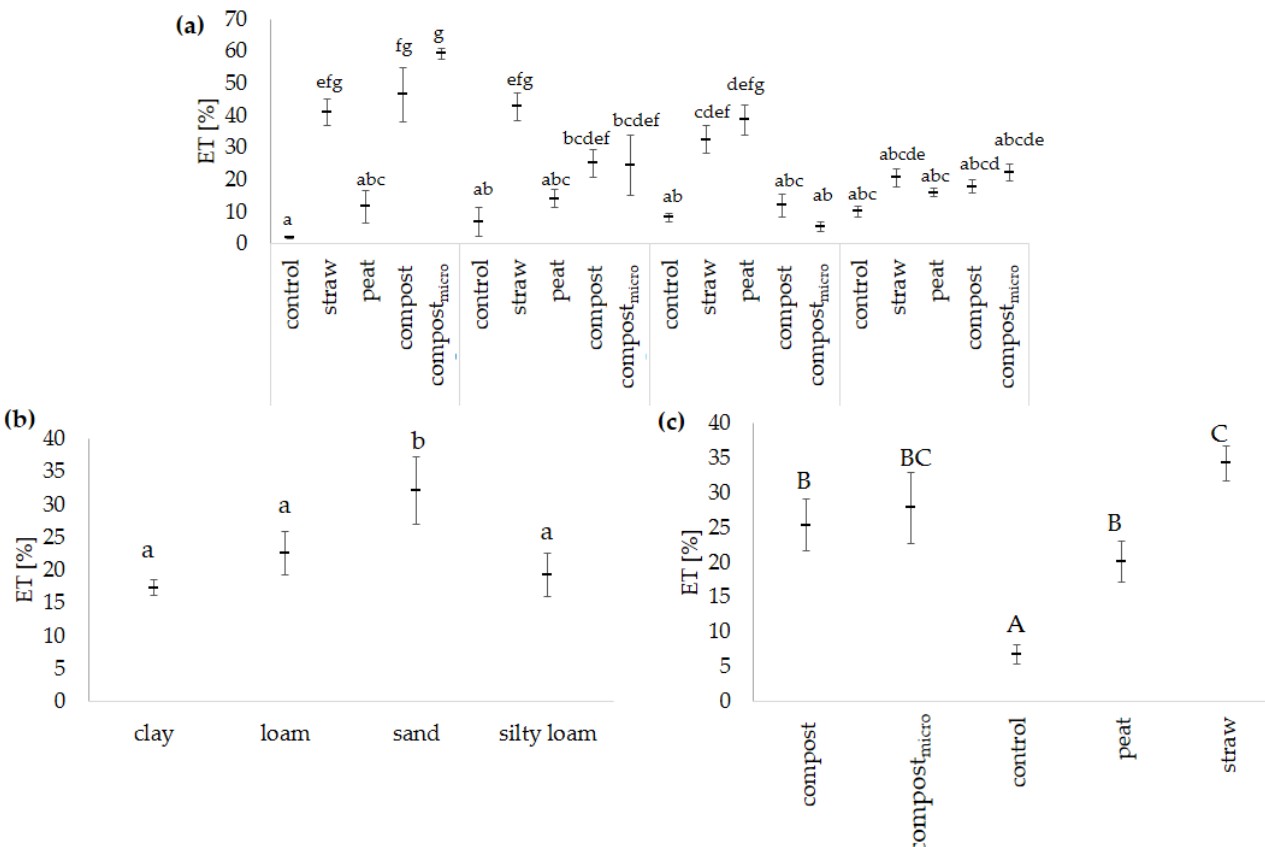

**Figure 3.** Earthworm traces (ET) in (**a**) all treatments, (**b**) by texture and (**c**) by addition. Different letters mean significant differences between (**a**) treatments, (**b**) texture and (**c**) additives.

The amount of $N_{intr}$ and $C_{intr}$ were correlated with dissolved organic carbon (DOC $p < 0.001$, $r = 0.425$ and $p < 0.05$, $r = 0.293$, for $N_{intr}$ and $C_{intr}$, respectively) and dissolved nitrogen (DON, $p < 0.001$, $r = 0.484$ and $p < 0.05$, $r = 0.284$ for $N_{intr}$ and $C_{intr}$, respectively) and microbial activity, expressed as microbial biomass carbon (MBC $p < 0.001$, $r = 0.563$ and $p < 0.01$, $r = 0.352$ for $N_{intr}$ and $C_{intr}$, respectively). Additionally, it was confirmed the correlation of $N_{intr}$, with microbial biomass nitrogen (MBN, $p < 0.01$, $r = 0.366$).

The soil texture and additives were factors affecting the microbial activity expressed by microbial biomass carbon and nitrogen. Treatments with compost and compost$_{micro}$ had the highest microbial activity; the lowest was in the control and treatments with straw (Table 1). The clay–compost treatment characterized significantly the highest content of MBC and MBN compared to other treatments. Additionally, the sand–compost treatment differed significantly from the sand control. High microbial activity (MBC and MBN) was accompanied by a higher amount of dissolved carbon and nitrogen (DOC and DON), easily available for microorganisms. These soil properties were correlated (DOC with MBN and MBC $p = 0.005$, $r = 0.397$ and $p = 0.000$, $r = 0.545$, respectively; and DON with MBN and MBC $p = 0.002$, $r = 0.440$ and $p = 0.003$, $r = 0.418$, respectively). The highest contents of DOC and DON were in silty loam–peat and clay–compost, respectively. The lowest contents were in sand control for DOC and sand–straw for DON (Table 1). Noteworthy is the percent of DOC in $C_{org}$—it was 0.6%, 0.8%, 1.5% and 4.0% in clay, silty loam, loam and sand, respectively, and 0.7%, 0.9%, 1.0%, 2.7% and 3.5% in peat, compost$_{micro}$, compost, straw and control, respectively.

The mean sieve size aggregate, based on 54 samples, was $1.58 \pm 0.12$ mm (the coefficient of variation 0.08). The size of aggregates for individual textures can be arranged in the descending order of loam > silty loam > clay > sand and were $1.64 \pm 0.10$ mm, $1.63 \pm 0.10$ mm, $1.53 \pm 0.11$ mm and $1.48 \pm 0.13$ mm, respectively (Table S3).

The stability of aggregates determined by the wet sieving method (WRI) varied and ranged from 0.05 in silty loam–peat to 0.89 in clay–compost$_{micro}$. Furthermore, the factors differentiating the WRI values were both the texture and the additive used. Significantly, the highest WRI values were in soils with clay texture, and the lowest were in silty loam and loamy soils. Among the organic additives, compost and compost with added microorganisms stimulated the formation of persistent aggregates the most. In loam–compost$_{micro}$, WRI was significantly higher than in loam–peat and loam–straw (Table 4).

**Table 4.** Aggregate stability in treatment soils, determined by the wet sieving method (WRI) and the laser diffraction method (ASI$_{LD}$). Different lower case letters mean significant differences for soil texture, and uppercase letter mean significant differences for additives.

| Texture Additive | WRI | | | | ASI$_{LD}$ | | | |
|---|---|---|---|---|---|---|---|---|
| | Sand | Loam | Silty Loam | Clay | Sand | Loam | Silty Loam | Clay |
| Control | * | 0.20 ± 0.14 ABCa | 0.11 ± 0.03 ABCa | 0.86 ± 0.06 ABCb | * | −1516.6 ± 1.4 Aa | −1543.1 ± 0.4 Aa | −990.8 ± 26.7 Ab |
| Straw | * | 0.10 ± 0.01 ABa | 0.16 ± 0.03 ABa | 0.86 ± 0.01 ABb | * | −1509.8 ± 1.1 Aa | −1538.0 ± 3.0 Aa | −989.0 ± 12.5 Ab |
| Peat | * | 0.11 ± 0.02 Aa | 0.05 ± 0.02 Aa | 0.85 ± 0.01 Ab | * | −1518.4 ± 1.8 Aa | −1541.3 ± 1.2 Aa | −936.6 ± 66.2 Ab |
| Compost | * | 0.23 ± 0.04 BCa | 0.18 ± 0.01 BCa | 0.88 ± 0.01 BCa | * | −1528.1 ± 4.0 Aa | −1541.9 ± 0.9 Aa | −939.5 ± 45.6 Ab |
| Compost$_{micro}$ | * | 0.27 ± 0.10 Ca | 0.20 ± 0.01 Ca | 0.89 ± 0.01 Cb | * | −1525.9 ± 3.0 Aa | −1537.7 ± 5.9 Aa | −992.2 ± 54.8 Ab |

* not enough aggregates.

The results obtained with the laser diffraction method (ASI$_{LD}$) confirmed the connection between aggregate stability and texture but did not confirm the connection of stability with additives. Significant the lowest ASI$_{LD}$ values were noted in the silty loam-control (−1543.1) and the highest in the clay–peat treatment (−936.6). Based on those results the highest water-stable aggregates occurred in clayey soils and the lowest in silty loam soils (Table 4).

Water stability of soil aggregates as measured by the laser diffraction method (ASI$_{LD}$) was strongly correlated with the water-resistant index determined by the wet sieving method (WRI). Due to the small number of sandy soil samples with measurable aggregates, these comparisons were only made for silty, clayey and loamy samples (Figure 4).

In the interpretation of relation were included first and second principal components (PC), because the first two axes explained 99.3% of the variability, and the assumption that a correlation above 0.5 is deemed important was made (Figure 5 and Table 5). The first PC was strongly correlated with four of the original variables (positively with C$_{intr}$, ASI$_{LD}$ and WRI, and negatively with HA). PC1 increased with increasing ASI$_{LD}$, WRI and C$_{intr}$ and decreased with DHA. PC1 was primarily a measure of aggregate stability (WRI and ADI$_{LD}$). PC2 increased with the increase of N$_{intr}$ and microbial activity (MBC and MBN) and dissolved components (DOC and DON) (Figure 3 and Table 5). The distribution of treatments in the ordination space (Figure 5) showed that soil texture was the grouping factor.

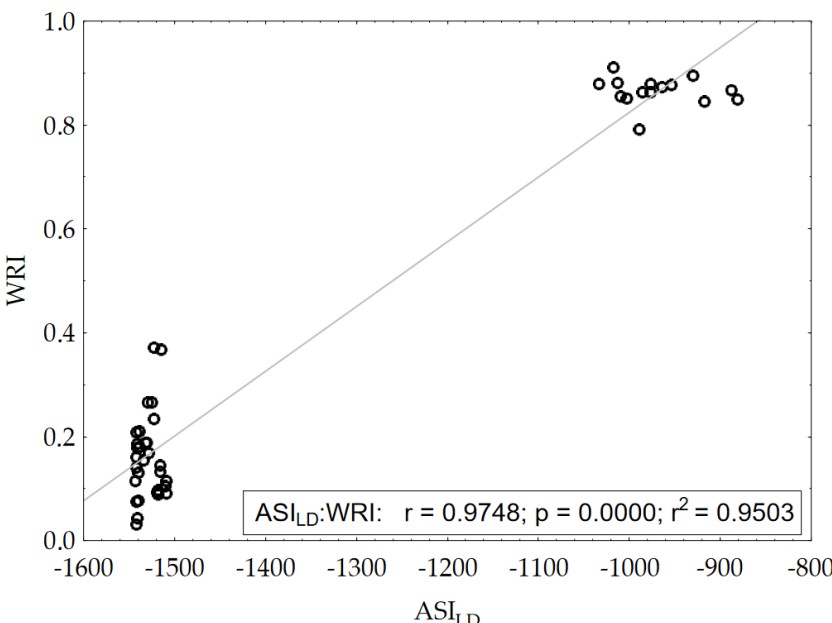

**Figure 4.** Comparison of the water-resistance index (WRI) and aggregate stability index (ASI$_{LD}$).

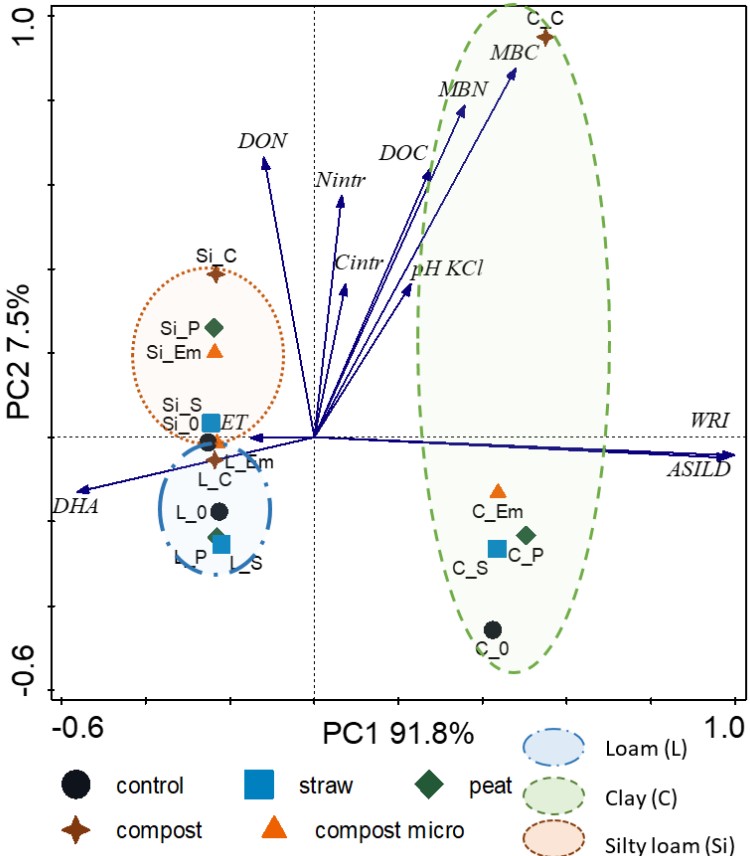

**Figure 5.** Principal component analysis with separation for texture and additives (abbreviations in formula a_b, where a is: L—silty loam, C—clay, Si—silt; and b is: C—compost, Em—compost$_{micro}$, S—straw, 0—control, P—peat, WRI—water-resistant index, ASI$_{LD}$—aggregate stability index, C$_{intr}$, N$_{intr}$—carbon and nitrogen introduced to the soil, MBC and MBN—microbial biomass carbon and nitrogen, DOC and DON—dissolved organic carbon and nitrogen, respectively, and DHA—dehydrogenase activity)

**Table 5.** Principal components for axis 1 and axis 2 in PCA analysis shown in Figure 5. Significantly correlated variables are bolded.

| Properties | Principal Components | |
|---|---|---|
| | **1** | **2** |
| $N_{intr}$ | 0.066 | 0.574 |
| $C_{intr}$ | 0.743 | 0.364 |
| $pH_{KCl}$ | 0.231 | 0.364 |
| DOC | 0.277 | 0.638 |
| DON | −0.120 | 0.664 |
| DHA | −0.562 | −0.130 |
| ET | −0.150 | −0.001 |
| MBC | 0.479 | 0.877 |
| MBN | 0.358 | 0.789 |
| $ASI_{LD}$ | 0.999 | −0.043 |
| WRI | 0.984 | −0.048 |

## 4. Discussion

In order to be able to properly interpret the obtained results, it is necessary to discuss the advantages and limitations of the measurement methods used in advance.

The measurement of ET was done by image analysis of the bottom surface photo. This method is commonly used in such experiments (e.g., [42,43]). However, it should be stated that the results obtained from this method should be treated rather as the measure of earthworm activity, not as the actual information. This is due to the fact that the 2D photo is made at the surface, which is close to the bottom of the container. The real movement of earthworms in the bulk of the soil is surely different. However, for the purpose of the investigations described in this work, the comparisons of the results obtained from such photos are reliable, because it can be assumed that the error is systematic. Each measurement was carried out in the container having the same geometry, and the volumes of soil, the numbers of earthworms, the procedure of photography and image analyses were the same in each replication.

The wet sieving method and calculation of WRI are also standard procedures commonly used (e.g., [31,44]). Each method also has sources of uncertainty, which are closely connected to the device used, the main ones being the duration of the measurement, stroke and frequency. However, since each measurement was carried out with a well-known device with the same (recommended by the producer) settings, the comparisons are reliable and repeatable.

The laser diffraction method is the relatively least frequently used measurement of aggregate stability. The results obtained by this method are strongly dependent on the device's settings and the procedure for determining the first measuring point [33]. However, with the given and validated settings, this method is sensitive and allows one to measure such aggregates for which the wet sieving method is not valid. As all measurements were carried out with the reproducible conditions, the obtained data are fully interpretable in the context of result comparisons.

Medium texture soil is more favorable to earthworms than clayey or sandy soils [45]. A comparison of clay loam and sandy loam soil with mineral fertilization or compost [46] confirmed that clay loam soil with compost has the most beneficial conditions for earthworms. The results only partly confirmed that relation—the lowest earthworm activity was in clayey soil, but the highest was in sandy soils (Figure 3b). However, low earthworm abundance in natural sandy soil is connected with water deficits, and in our experiment, water conditions were optimal, which may indirectly demonstrate that the low presence of earthworms is related to soil moisture and not strictly related to a sandy texture. In the experiment, a higher activity of earthworms in treatments with all additives compared to the controls was observed (Figure 3c). The highest percentage of earthworm tracks (ET) were in soils with straw, but this did not correspond with the amount of carbon introduced to the soil (Table 1) or the stability of the aggregates (Table 3).

In treatments with straw, which had the highest ETs, the content of $C_{intr}$ was the lowest. This may be connected with microbial activity (MBC and MBN), which was also the lowest in the straw treatment. Moody et al. [47] emphasized that earthworm/microbial interactions support straw decomposition. Enhancement of microbial activity may accelerate organic matter mineralization, which is not beneficial for C stabilization in soil. Wu et al. [48] noted that earthworms in treatments with straw supported carbon storage in macroaggregates (>0.25 mm); however, during their 40-day experiment, only 3.8% of added straw C was assimilated by earthworms. For comparison, in our six-month experiment, less than 8.5% of the organic carbon from the straw was incorporated into the mineral part of the soil.

The PCA diagram shows that texture was the most differentiating factor considering the investigated soil properties. The activity of dehydrogenases, like the earthworm traces, was one of the parameters that was negatively correlated with the stability of aggregates, which can be related to the fact that DHA is an indicator of the soil microorganisms' respiratory metabolism [49], and therefore excessive aggregation and high content of clay may have a destimulating effect. Similar results were obtained for ET, but as shown in the PCA (Figure 5), it is the parameter that influenced the position in the ordinance to a small extent (short arrow).

Soil organic carbon, which may be incorporated into the soil in the form of organic–mineral colloids, is an essential element in the balance of carbon in nature. Among the tested additives, organic carbon from compost, peat and compost with active bacteria cultures was in the largest component incorporated to fine earth particles (about 36–48%). The addition of compost with earthworms increased the content of labile organic matter [50], which was confirmed in our results; the highest content of dissolved organic carbon (DOC) occurred in treatments with compost (Table 1). The presented results confirm the conclusion of Lapied et al. [46], that inputs of organic residues (e.g., urban compost) are beneficial for earthworm activity and soil quality (e.g., aggregate stability). However, the results do not support the findings of Zhang et al. [51], who demonstrated that the application of straw has positive effects on aggregate stability and the content of organic carbon. The differences between the conclusions of Zhang et al. [51] and the presented results may be caused by the fact that Zhang et al. [51] used mineral fertilizers in the experiment (e.g., nitrogen), and in the conducted research only introduced straw. Józefowska et al. [52] concluded that nitrogen is a factor that limits the growth of microorganisms. This is also noticeable in the present study, e.g., the straw treatments had the lowest DON and MBN content and low $N_{intr}$ content. Summing up, based on a six-month experiment with optimal moisture conditions, straw alone without additional nitrogen fertilization is not conducive to the development of stable aggregates and incorporation of organic carbon in fine earth particles. Nitrogen content may be a limiting factor for microorganisms, but soil fauna may be as well. Ponge [53] emphasized that the mineralization and nitrogen uptake by plants and microorganisms is very slow or even impossible when the C:N ratio is high (more than 33). In such soils, the faunal activity and decomposition process is low, and litter accumulates on the soil surface.

Opinions on the formation of aggregates by earthworms differ. Many authors [54,55] report that earthworms are structure-forming, e.g., a worm cast structure is formed during the passage of soil through their digestive tract, but some authors [56] claim that bioturbation causes aggregate disruption. Based on the results, the connection between earthworms and aggregate formation cannot be directly confirmed, but it has shown that organic additives support the activity of earthworms and microorganisms and that the stability of soil aggregates is also dependent on them.

Soil organic matter and texture are the major factors that affect aggregate stability [57]. The research presented here confirmed that mainly soil texture but also the content of organic carbon are related to the stability of soil aggregates (Table 2). Many authors emphasized the significant role soil organic matter and texture (clay content) play in the formation and stabilization of aggregates [58–61]. In our results, the most stable aggregates

occurred in clayey soils, then loamy and silty loam, and the weakest aggregates occurred in sandy soils. According to Franzluebbersl [62] and Hassink [63], the soil clay content influences the degree of aggregation due to the highly reactive surface of clay binding with the negative surface charges of soil organic matter.

According to Zhang et al. [64], soil organic matter has a slight impact on the stability of soil aggregates and is not a direct influencing factor, but can indirectly improve it, shaping the microbial activity (MBC and MBN) in the soil. Compost and compost with microorganisms support biological activity, and the stability of the aggregates was in line with Adugna [65] and Lapied et al. [46], who note that the introduction of compost into the soil improves the aggregation process and the stability of the formed aggregates.

In the Introduction, it was emphasized that stable soil aggregates are an important element of soils. Several methods are used to assess the stability of soil aggregates and there are several studies comparing those methods (e.g., [8,32,33,54]). Almajmaie et al. [31] compared rainfall simulation, wet sieving, ultrasonic vibration and clay dispersion and concluded that so far no method is universal for all soils and conditions. Therefore, research is still needed towards a unified method that will be able to provide information about the stability of all types of aggregates. In this study, two methods, the well-known wet sieving and a relatively new one, i.e., the disintegration of aggregates using laser diffraction were compared.

Drawbacks of this research were the lack of medium-stable aggregates (Figure 4) and the lack of a small number of soil aggregates with a sandy texture. As is known, sandy soils often form unstable aggregates or even do not develop an aggregate structure and are split-grained. This issue requires a separate study and the collection of an appropriate number of sandy soil samples, e.g., from natural soil profiles. Based on the observations made on those aggregates with a sand texture that were successfully tested, it was noticed that in the wet sieving method, sand above 0.25 mm (this is the sieve diameter in the wet sieving method) may have "pretend" stable aggregates, which disturbs the results. Therefore, it is important to apply sand corrections when using the wet screening method. This fact was emphasized by Seybold and Herrick [66]. Similar observations can be made by analyzing the aggregate disintegration graph (Figure 2). Aggregates with sandy texture do not break down into particles below 400 μm. In the $ASI_{LD}$ method, such a correction is not required, because in this method, the breakdown of aggregates after 1 min is compared (Figure 2).

Excluding sandy soils from the comparison (due to the small number of repetitions), it can be concluded that the two tested methods give comparable results (Figure 4). However, the correlation between two groups of points far from each other necessarily generates a strong correlation, and intermediate points would be essential to validate such a correlation, but in the experiment such results were not obtained. However, on the basis of Figure 4, it can be seen that WRI differentiates the stability of weak aggregates well, while the $ASI_{LD}$ differentiates well the aggregates with strong stability. Those results are partly in line with Bieganowski et al. [33], who found that the $ASI_{LD}$ and WRI methods are comparable, especially for weak and moderate aggregates. Our results showed that the $ASI_{LD}$ method is proper also for strong aggregates.

## 5. Conclusions

Earthworm activity is one of the factors favoring the formation of aggregates. The stability of these aggregates depends on the soil texture (the strongest aggregates are created in clay soils) and organic additives introduced into the soil. In terms of the investigated organic additives, in combination with earthworm activity the efficiency of aggregate creation was as follows: compost with active bacteria, compost, peat and straw. This efficiency was caused by additional carbon and nitrogen availability because of the additional microbial activity. Based on our results, it can be stated that the stability of aggregates is in part due to soil texture and the content of organic matter, which stimulates the development of microorganisms. The study indicates that among the tested additives,

regardless of the soil texture, compost and compost with bacterial cultures most stimulated the development of stable soil aggregates and the binding of organic matter (expressed by the amount of introduced carbon and nitrogen). Earthworms alone, without the addition of an organic additive that delivers available organic carbon for microorganisms, do not form permanent aggregates, as shown by the control and the treatment with straw. In soil with stable aggregates, carbon stabilization can take place (lower DOC:$C_{org}$ ratio).

Two methods (i.e., wet sieving and laser diffractometry) to measure aggregate stability were comparable for silty, clayey and loamy soils. Sandy soils need further investigation because in the wet sieving method, sand fractions higher than 0.25 mm may provide false or spurious measures of stable aggregates.

**Supplementary Materials:** The following are available online at https://www.mdpi.com/2073-4395/11/3/421/s1, Table S1: Location and selected properties of the soils used in the experiment (WHC—water holding capacity), Table S2: Carbon content in additives and their doses in the individual treatments, Table S3: The diameter of soil aggregates, earthworms' traces (ET) and the dissolved organic carbon to organic carbon ratio (DOC:Corg) (SD—standard deviation).

**Author Contributions:** Conceptualization, A.J. and A.B.; methodology and investigation, A.J., K.W., J.S., A.S., M.R.; writing—original draft preparation, A.J.; writing—review and editing, A.J., A.B., M.R., T.Z. K.W., J.S., A.S.; visualization, A.J.; funding acquisition, A.J. All authors have read and agreed to the published version of the manuscript.

**Funding:** This research was funded by THE NATIONAL SCIENCE CENTRE, Poland grant number 2017/01/X/ST10/00777.

**Data Availability Statement:** The raw data of results presented in this study are available on request from the corresponding author.

**Acknowledgments:** We are grateful to Bartłomiej Kajdas, PhD for technical support in image analysis, and SUEZ Południe Sp. z o.o. and ProBiotics™ Polska for providing compost and probiotics, respectively. We wish to thank the Cambridge proofreading team for language corrections. We wish to thank the Cambridge proofreading team for language corrections. We would like to thank the reviewers for their comments and efforts towards improving our manuscript.

**Conflicts of Interest:** The authors declare no conflict of interest.

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
