# Peer review of "Stability of Aggregates Made by Earthworms in Soils with Organic Additives"

_agronomy, doi:10.3390/agronomy11030421_

Round 1

Reviewer 1 Report

Very good paper.

Please, stay in good condition and continue with investigations, especilay with medium-stable aggregates in the sandy soils (samples from natural soil profiles).

Best Regards

Author Response

Thank you for your positive review. 

Reviewer 2 Report

Honestly, I found this document very interesting to increase the knloledge on soil agregates formation.

 It is the fist time I found the manuscript without any objection and easy Reading, at the time that it reflects the corretation of many variables with the soil agregate stability.

As I said, the manuscript is well explained and it covers all the needed part a paper should have. However, the discusión is crucial and very particular in this context. It is extended and reflect waht many other authors had previously developed and stated in the same context.

Therefore, I have to recognise this papaer could be accepted as it is now.

Author Response

Thank you for your positive review.

Reviewer 3 Report

Paper entitled „Stability of aggregates made by earthworms in soils with organic additives“ deals with investigation of two factors (soil texture x organic additives) on soil aggregation. It’s an interesting paper and fits the scope of the Journal. Before it could be recommended for publication authors should consider next things.

The introduction should contain previous knowledge of aggregation and cementation of aggregates in sandy and clay soils. Moreover, few sentences should be added on type of organic matter and their impact on soil aggregation. At the end of introduction, the hypothesis should me added. And aims as well.

Materials and methods section should have few upgrades. Please state soil types and mark the horizons from where you took the samples according the IUSS WRB. Please state the doses of organic amendments used in each soil type.

Line 74: Add the commercial name, manufacturer and state of the production. Same for the lines 107 and 109.

Fig 1. Please use standardise arrows and font in this figure. Please avoid hand drawings if possible.

Line 111-112. Please add the shaker type and duration of shaking.

Figure 2. Please remove vertical lines and outline.

Tell to the readers do you have normal distribution datasets to justify the used statistical procedures.

Results: You cannot call first the Table 2 before Table 1 in the text. Please adjust.

The results should be described scientifically. Which treatment was significantly higher or lower than other, etc. Please perform this in whole section. In Tables 1 and 2 please explain what different lower letters represent. Tables and figures should be self-explanatory.

Figure 3. Please remove gridlines. In this property you miss the statistical results. Please update.

Line 208. Add comma

Line 203-206. Please state the source where this data was presented.

Please add full word before abbreviation in title of table 3.

Something is strange in the statistical results. Although reader cannot know what different letter means because nowhere is explained, I notice the repetitions of some letters which are strange. Please use the lowercase letter to explain the difference between factor 1 (texture) and uppercase letter to explain the difference between factor 2 (amendment). This should be more visible and more readable. Please do this for all tables.

Figure 4. remove gridlines and outline for statistical results in figure.

Table 4. Please mark significant interrelations between variables. Please present which variables were significantly correlated (positively and negatively) in PC1 and which in PC2. I assume that significant ones are mentioned ones in lines 233-238.

Line 233-234. State which ones.

Discussion part usually discuss the properties in an order how are presented in results section. I see that authors use different strategy and highlight most important findings and then continue with the others. This could be proper as well. This part is properly written and have some merit. Some of the properties, however, missing into discussion (e.g. DON, DHA). I suggest that add them in discussion part as well. Moreover, the interrelations between variables obtain by PCA also did not highlighted in Discussion part. In this section please take care for style of writing: e.g. Line 258. “go hand in hand” …. please avoid such terms. Line 277. Please avoid using “We”….check entire manuscript.

Line 286-287. Please state this together with notification of duration of the experiment and environmental conditions during experiment. You cannot know which results you can measure after e.g. 2 years when N depression pass.

Conclusion is supported by results. Although one take-home message should be added for practitioners: e.g. the best organic source for each soil texture type.

Reviewer 4 Report

Génral comments :

The granulometry of the soils is not at all sufficient to present the soils. 
Lines 76 and 77, the carbon content should be expressed in a unit of the international system.
Nothing is said about the initial state of aggregation of the soils used before their removal, in situ, nor how organic matter was added.
During the 6-month experiment, organic matter can evolve differently even without interaction with earthworms.
The control is not really correct because the behavior, the functioning and even the survival of earthworms without the addition of organic matter is not similar. Like a crop receiving water but no nutrients. The effect of worms could have been measured with a worm-free control.

Part Results
line 145 to 157: these items are rather material and method.
The method of visible earthworm traces (ET) is not sufficiently developed. No information is given on the quality or accuracy of this measurement. The standard deviations in Figure 3 are all the same (ca. 18%), which suggests that they were set in order to calculate statistical tests. The significance of these SDs is also not explained. Moreover, the results in Figure 3 are not presented in detail.
"Furthermore, the factors differentiating the WRI values were both the texture and the additive used." 
Table 3 uses repetition to determine if the differences are significant. This is true, but the measurement itself is tainted by its own error that should be added to the standard deviations of the replicates.
WRI and ASI never show significant differences between the control and the organic input modalities regardless of the type of texture. 
Figure 4: the correlation between two groups of points far from each other necessarily generates a strong correlation. However, intermediate points are essential to validate such a correlation. 

Globally the results are very unconvincing (low robustness of the statistical tests, weak quality of the methods used). Next, the discussion is very general. It makes very little use of the results obtained; it mainly analyzes the literature on the characteristics evaluated. 

Round 2

Reviewer 3 Report

The authors positively response on all my open questions. I suggest this work to be considered for publication. Please just improve few minor things: in Table S2 instead "IUSS WRB" write "Soil Type". In statistical procedures provide to readers the variables which were transformed before ANOVA test.

Kind regards

Reviewer 4 Report

ok with these corrections.